



# Fixed photogrammetric systems for natural hazard monitoring with high spatio-temporal resolution

Xabier Blanch[1,2], Marta Guinau[2], Anette Eltner[1], Antonio Abellan[3]

[1] Institute of Photogrammetry and Remote Sensing, Technische Universität Dresden, 01062 Dresden, Germany
[2] RISKNAT Research Group, GEOMODELS Research Institute, Universitat de Barcelona, 08028 Barcelona, Spain
[3] Center for Research on the Alpine Environment (CREALP), Sion, CH1950 Valais, Switzerland

*Correspondence to*: Xabier Blanch (xabier.blanch@tu-dresden.de)

**Abstract:** In this publication we address the lack of knowledge in the design and construction of photogrammetric systems for high spatio-temporal resolution rockfall monitoring. Accordingly, we provide in-depth information on the components,
assembly instructions, and programming codes required to build them, making them accessible to researchers from different disciplines who are interested in 3D change-detection monitoring. Each system comprises five photographic modules and a wireless transmission system for real-time image transfer. As an alternative to LiDAR (Light Detection and Ranging), high-end digital cameras offer a simpler and more cost-effective solution for the generation of 3D models, especially in fixed time-lapse monitoring systems. The acquired images, in combination with algorithms that allow the creation of improved 3D
models, offer change detection performance comparable to LiDAR. We showcase the usefulness of our approach by presenting real-world applications in the field of geohazards monitoring. Our findings highlight the potential of our method to detect pre-failure deformation and identify rockfalls with a theoretical change-detection threshold of only 3-4 cm, thereby demonstrating the potential to achieve similar accuracies to LiDAR but at a much lower cost. Furthermore, thanks to the higher data acquisition frequency, the results show how the overlap of events that leads to an erroneous interpretation of the behaviour of
the active area is minimized, allowing, for example, more accurate correlations between weather conditions and rockfall activity.

## 1 Introduction

Close-range remote sensing monitoring is a powerful tool for understanding and capturing natural environments, especially used in the field of natural hazards (Giordan et al., 2022). Its ability to provide detailed and accurate data in real time makes it
a key monitoring tool in natural science research. This approach applied in the investigation, modelling and understanding of geological hazards is of utmost importance for our society, with a wide range of applications including the investigation of landslide phenomena using different 3D surveying techniques such as traditional photogrammetry (Gili et al., 2021) or LiDAR (Jaboyedoff et al., 2012) to advanced systems that can detect volumetric changes (Kromer et al., 2018) or pre-failure deformation (Royán et al., 2015) on active rockfalls.

Among the different close-range monitoring strategies that can process both 2D and 3D information, digital photogrammetry has played a crucial role in natural hazards monitoring. Its popularity is reflected in the large number of publications that use photographic images to make 2D measurements or obtain point clouds for 3D analysis (Anderson et al., 2019). As an example, one evidence of this role is the widespread use of really low-cost, easy-to-install cameras capturing time-lapse images, such as a) wild cameras for glacier front monitoring (Mallalieu et al., 2017) or b) Raspberry Pi cameras to monitor rock walls (Blanch
et al., 2020, Santise et al., 2017). While these cameras offer a novel, ready-to-use and quite cost-effective resource for 2D and 3D monitoring, their limitations, such as low-resolution images and poor geometric quality, must be taken into account. In consequence, there is a motivation to explore photogrammetric systems for 3D monitoring using digital cameras whose price, size and weight is not a constraint.

The use of high-definition photogrammetric systems for 3D monitoring can yield superior results in comparison to the low-
cost systems. By deploying fixed time-lapse systems equipped with high-quality cameras and lenses, researchers can acquire 3D models that can be compared to those captured by other high-end monitoring devices, albeit at a lower cost. High-end cameras for digital photogrammetry also offer several advantages, including lower infrastructure costs, ease of installation in remote locations, and a low learning curve, contributing to its democratization as a monitoring (Eltner and Sofia, 2020; Smith et al., 2015; Westoby et al., 2012). As an example of its widespread use, high-end photogrammetric systems are employed to
analyze and monitor objects and surfaces in a wide variety of fields such as building structures (e.g., Artese et al., 2016; Bartonek and Buday, 2020; Castellazzi et al., 2015; Meidow et al., 2018) or natural surfaces (e.g., Eltner et al., 2016; Nesbit and Hugenholtz, 2019; Westoby et al., 2012).

When high-end cameras are installed as a single system, the camera can capture time-lapse image sequences from a single location, which have been widely utilized in various scientific studies that work with 2D data. Evidence of the effectiveness
of such approach can be found in numerous publications like: a) the use of a camera for semi-automatic flood monitoring of a


proglacial outwash plain (Hiller et al., 2022), b) the installation of a camera to monitor the active areas on Alpin terrain (Hendrickx et al., 2022; Travelletti et al., 2012), c) the tracking of glaciers with a monoscopic camera (Hadhri et al., 2019; Schwalbe and Maas, 2017) or d) the monitoring of processes in glacial fronts (How et al., 2019; Lenzano et al., 2014).

Integrating a second photographic module enables the creation of a basic photogrammetric strategy, i.e., stereo-pair time-lapse.
3D information can be derived from the 2D images captured utilizing photogrammetric processing algorithms. Thereby, 3D models can be reconstructed providing a more comprehensive understanding of the monitored subject's spatial characteristics. The advantages of this strategy are illustrated in the various studies covering different fields of geosciences: a) the use of oblique stereo-pair time-lapse imagery to characterize active lava flows (James and Robson, 2014), b) the application of two cameras fixed on a monorail to perform interior tunnel monitoring (Attard et al., 2018), c) the assessment of a two-camera
system for fluvial surface monitoring (Bertin et al., 2015), d) the installation of two cameras for rockfall monitoring (Giacomini et al., 2020; Roncella et al., 2021) or the setup of two cameras to analyse glacial oscillations in the Glaciar Perito Moreno (Lenzano et al., 2018).

As a result of the performance of Structure-from-Motion and Multi View Stereo algorithms (SfM-MVS), which has been extensively detailed in the literature (e.g., Eltner and Sofia, 2020; Iglhaut et al., 2019; Mike R. James and Robson, 2014;
Westoby et al., 2012), it is becoming increasingly common to observe the deployment of more comprehensive photogrammetric systems for natural hazard monitoring purposes. Notable examples of such systems, resembling the approach proposed in this publication, are the five cameras installed at a distance of 80 meters to a slope for rockfall monitoring (Kromer et al., 2019), this system had data transfer capabilities and could achieve a level of detection between 0.02 to 0.03 m. A similar approach was recently conducted by Núñez-Andrés et al. (2023) with an installation of three cameras in front of the Castellfollit de la Roca cliff (NE Spain) with the aim to identify pre-failure deformations. Finally, Kneib et al., (2022) install two sets of
cameras (12 in total) in debris-covered glaciers in High Mountain Asia (China) to analyse the melt rates on a sub-seasonal basis.

However, these publications did not provide comprehensive descriptions of the construction of the high-end photogrammetric systems, the products used, and the codes developed for its operation. Moreover, not all of these systems were capable of
transmitting data remotely. Sometimes they are controlled by basic intervalometers, or they are installed in urban environments with access to the power grid. These monitoring systems can be implemented using a variety of methods and materials, including expensive commercial systems. But this study focuses on the systems that are installed on a fixed or semi-permanent basis and require custom-built enclosures, batteries, and control systems, tailored to the particular needs of the research, and which can be built DYI (do it yourself).

## 1.1 High-End Photogrammetry vs LiDAR

In the field of geosciences, LiDAR has become a popular method for 3D surface monitoring (Petrie and Toth, 2008) due to its ability to capture 3D models from long distances and at high speeds. Leading to significant advancements, for example, in the characterization of rockfalls (Abellán et al., 2006; Royán et al., 2014) or in a large landslide monitoring (Jaboyedoff et al., 2012) using high frequency data acquisition. These advances, coupled with the high quality of the data, especially considering
the spatially consistent error behaviour, has resulted in a steady increase in the use of LiDAR in geosciences (Guzzetti et al., 1999; McKean and Roering, 2004; Buckley et al., 2008; Tarolli, 2014; Tonini and Abellan, 2014).

Despite its many advantages, LiDAR technology also presents certain limitations such as high infrastructure costs, difficulties in the use in remote locations, and high fixed installation costs (Cook, 2017; Smith et al., 2015). These challenges make LiDAR implementation incompatible for a massive deployment in the field of natural hazards monitoring (Cook, 2017; Sturzenegger
et al., 2007; Wilkinson et al., 2016) where data collection needs to be repeated periodically.

Many studies have attempted to assess the accuracy of digital photogrammetry in geosciences (Brunier et al., 2016; Cook, 2017; Eltner & Schneider, 2015; James and Quinton, 2014; Luhmann et al., 2016; Buckley et al., 2008; Smith et al., 2015). Thanks to advances in digital photogrammetry and computer vision, the image-based 3D reconstruction results can now be compared to those obtained by LiDAR (Cook and Dietze, 2019; James and Robson, 2012; Rowley et al., 2020; Stumpf et al.,
2015). Furthermore, the effectiveness of high-end photogrammetry for fixed time-lapse monitoring can be comparable to those obtained using LiDAR technology (Kromer et al., 2019), which highlights the potential of ad hoc fixed photogrammetric systems as a viable alternative for real-time natural hazards monitoring.

However, there is a requirement for further progression in the realm of high-resolution photogrammetric systems. In particular, there is a pressing need to make the installation more robust and improve the reliability of these systems in order to guarantee
time-lapse acquisition as well as to increase real-time data capture and transmission to support autonomous data collection in harsh environments. Such advancements, along with the integration of advancements in point cloud and SfM-MVS processing (Blanch et al., 2020; Brodu and Lague, 2012; Feurer and Vinatier, 2018; Gómez-Gutiérrez et al., 2015), can enable these systems to carry out real long-term natural hazards monitoring with accuracy and reliability comparable to LiDAR, but at a significantly lower cost.




Last, these LiDAR vs digital photogrammetry comparisons cannot be generalized because are dependent on factors such as the camera used, the number of photographs taken, and the distance and shape of the object (Eltner et al., 2016). Due to these complex uncertainties, one potential way to analyse the performance of the 3D measurement methods consists of making measurements of the same object with the acquisition instrument emplaced at the same distance. Thus, this study compares change-detection metrics using LiDAR instrumentation (Abellán et al. 2010) and the high-resolution photogrammetric system
described in this publication. Both data acquisitions have been carried out on the Puigcercós study area (NE, Spain) with the LiDAR and the cameras placed at the same distance from the escarpment.

The subsequent section delves into the specific requirements for autonomous photogrammetric systems, providing a comprehensive overview of their components and assembly. A practical demonstration of their application in real-world scenarios is then presented, along with the resultant data and an evaluation of its reliability for rockfall monitoring. Finally,
insightful discussions on potential areas of improvement and the mitigation of common errors are provided, with the goal of advancing fixed time-lapse photogrammetric systems for change-detection monitoring.

With the aim of assisting researchers from various disciplines who may not have experience in photography, electronics, or computing, this publication includes in an open distribution of the assembly instructions, programming codes, and components used to successfully develop advanced high-resolution photogrammetric systems. There is a large opportunity to empower
society in acquiring more data and gaining a deeper understanding of its environment by making this information freely accessible. This has not only the potential to enhance the knowledge across disciplines, but also increase the overall safety and security of society by providing a greater awareness of its environment and its hazard, regardless of society's economic resources.

## 2 Material and methods

### 2.1 Photogrammetric system

There is no consensus in the literature on the definition of a photogrammetric system. The works shown above use different photographic modules to obtain time-lapse images from which 3D models can be reconstructed. Each of these systems has its own peculiarities and is adapted to the specific research task. It is therefore difficult to homogenise the basic characteristics that define a photogrammetric system versus a simple cameras array.

How we see it, a photogrammetric system is described as a set of synchronised cameras able to simultaneously capture images of an object aiming to generate a 3D reconstruction of the reality. The autonomous system described in this work is characterized by its ability to reliably monitor remote areas, for a long-time period and without the need of on-site intervention. The photogrammetric system consists of a combination of basic components, including a set of imaging devices comprising of a lens and sensor, a control unit for data acquisition, a robust network infrastructure for data transmission, and a self-
sufficient power supply to ensure continuous operation in remote locations. The development of these systems is based on three basic principles:

**i) On demand**. The programming/programming of the control units must allow full control of the shooting times and modes of the camera. (e.g., demand a burst of 15 photographs every five hours during the day). Note that systems with an external trigger, e.g., associated with a rain sensor, humidity sensor or seismic sensor should also be considered programmable because
they can acquire the data on demand, too (e.g., camera triggered by a change in infrared levels in Hereward et al., 2021).

**ii) Data transfer**. Due to the large amount of data that can be acquired with these systems, the photogrammetric systems should be remotely accessible to facilitate data transfer in both directions, i.e., capable of transmitting data as well as receiving basic programming instructions. In addition, IoT-based approaches where processing is done on-site and only small data amount, e.g., a result or alarm, is transmitted, are also considered as "data transfer" devices.

**iii) Stand-alone power system**. Fixed monitoring systems can be supplemented with appropriate power systems that allow not only a high degree of autonomy but also the ability to be self-sufficient from a power consumption standpoint. Thanks to the use of batteries, solar panels or other alternative power sources, it is now possible to install photogrammetric systems on remote location without access to conventional electrical infrastructure. Additionally, the use of very low energy consumption systems that allow several months up to years of operation are also considered.

Although, much cheaper photogrammetric systems may be available (based on low low-cost sensors) the design presented here is also referred to as low-cost or low budget according to the general terminology used in the literature (e.g., Anderson et al., 2019; Gabrieli et al., 2016; Giordan et al., 2016; Khan et al., 2021). This is because the final cost of the system is largely controlled by the cost of the camera and lens, which can be easily adapted to the budget and is much lower than other systems such as LiDAR, Radar, or total station (Smith et al., 2015).



### 2.2 Components of the photogrammetric systems

Several key components are required to create a photogrammetric system. On the one hand, the photographic modules discussed in this study, which contain the cameras, the control units and the electronics that enable an advanced acquisition system. On the other hand, the system of data transmission and the real-time photogrammetric processing workflow (Blanch et al., 2021). **Figure S1** in the supplementary material shows in detail the materials used for the development of the photogrammetric systems. In this study we share the knowledge and experience gained from systems installed at the Puigcercós cliff (NE Spain), located in the Origens UNESCO Global Geopark, and at the Alhambra de Granada (S Spain), an UNESCO World Heritage place. Both are protected areas of high cultural value that present rockfall activity.

The purpose of giving full details on the different elements that make up the photogrammetric system (i.e., brands and models), including the explanation of their assemblage and functioning, is to provide specific information to non-experts, who would like to setup their own autonomous 3D monitoring system. The decision to expose the components in detail is associated with the possibility of sharing the codes (software) and making them fully functional – i.e., plug&play - with the provided hardware. The codes can be found in open-source format in the first author GitHub repository: https://github.com/xabierblanch/DSLR-System.

#### 2.2.1 Photographic sensor and lens

Three different camera models and lenses have been tested. In the Puigcercós study site, three photographic modules use a full-frame ($35.9 \times 24$ mm) mirrorless Sony Alpha 7R III with a resolution of 42.4 MPx (pixel pitch of 4.51 µm). These cameras are equipped with a 35 mm f/2.8 lens. Two other photographic modules use a Canon 77D DSLR camera with an APS-C sensor (22.3 x 14.9 mm) and a resolution of 24.2 MPx (3.72 µm pixel pitch). The Canon cameras are equipped with a 24 mm f/2.8 pancake lens. Using a 24 mm lens on a cropped APS-C sensor produces approximately the same field of view as a 35 mm full-frame lens (**Figure S1 a and b**).

At the Alhambra de Granada study site, all five photographic modules were set up with the Nikon D610 DSLR full-frame camera ($35.9 \times 24$ mm) with a resolution of 24.3 MPx (5.95 µm pixel pitch). This publication also includes some preliminary evaluations with the Nikon Z5, a full-frame ($35.9 \times 24$ mm) mirrorless camera with 24 MPx resolution (5.97 µm pixel pitch).

#### 2.2.2 Control Unit and Real Time Clock – RTC

A microcomputer is used as control system of the photographic modules. The unit control is a Raspberry Pi 3 Model B+ (Raspberry Pi Foundation, 2016) (**Figure S1c**). It operates the system, controls image capture, transmits data and manages the camera's battery. A commercial Witty Pi 3 board from UUGear is used as a real-time clock and power management system for the Raspberry Pi 3 (**Figure S1d**). Specifically, the Witty Pi board allows to interrupt the activation of the system shutdown through the GPIO (General Purpose Input/Output) pinouts of the Raspberry Pi. This function is used in our design to allow remote access, without time limitation, to enable code modifications and upgrades of the system.

#### 2.2.3 Network connection system:

The photogrammetric system includes a network infrastructure that allows images to be uploaded to a server. The solution provided in the repository codes is based on the Dropbox data transfer system. However, the same solution can be applied to private servers or other proprietary and open-source cloud systems (**Figure S1e**). In addition, a connectivity unit is used in the Puigcercós (NE, Spain) and Alhambra de Granada (S, Spain) study areas with dedicated hardware to provide 4G connectivity to the study area. The main component of this system is the router with 4G capabilities that converts the phone signal into Wi-Fi. The router is powered by a battery and a solar panel (in the same way as the photographic modules). A 12V timer is in charge of switching the router on and off to prevent wasting energy during periods of inactivity.

The system is modular and easily scalable. In the case of the Puigcercós connectivity module, the default omnidirectional antennas of the modem have been replaced by longer range antennas. Both to receive the 4G signal (directional panel antenna) and to send the Wi-Fi signal to the photographic modules (semi-directional antenna). For the 4G antenna it is recommended to use panel antennas that have a highly concentrated radiation pattern at a single point. With a precise installation this antenna can be oriented in the direction of the nearest telephone antenna, optimising the radiation pattern to obtain the highest possible energy. **Figure S2** in the supplementary material shows the assembly scheme of the connectivity system, and **Figure S3** shows some images of the connectivity modules installed in the Puigcercós study site and the Alhambra in Granada.

#### 2.2.4 Power supply and electrical components

The system employs 12 V batteries to power its components (**Figure S1f**), as they require a higher voltage and consume more power than very low-cost alternatives which often use small batteries (1.5 V) or power banks (5 V). The sealed lead AGM




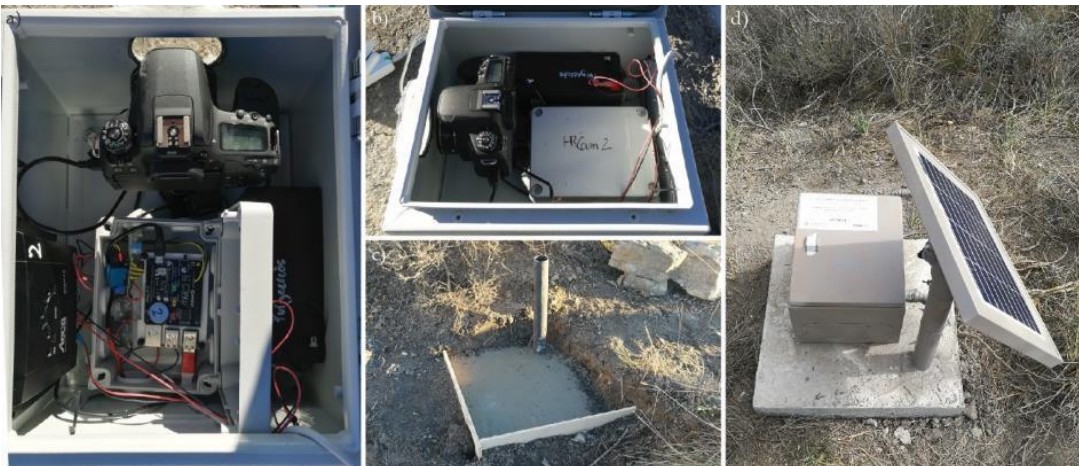

**Figure 1. Images of the photogrammetric systems installed in the Puigcercós site: a and b) Images showing the inside of the photogrammetric unit with part of the connections shown in Figure 3; c) Construction of the concrete base and installation of the PVC pipe for the installation of the photographic units and d) external view of the installed system.**

batteries used in the system have been proven to be reliable and durable, with capacities ranging from 7 Ah to 12 Ah. Solar
panels of 10 W or 20 W have been utilized to power the batteries (**Figure S1g**). The panels must be connected to a solar charge controller (**Figure S1h**), which manages the battery's charging and discharging processes.

The designed system is characterised by the need for small electronic elements that are responsible for adapting the power supply that reaches all the devices. Mainly, two systems are needed: The DC-DC voltage converter is responsible for modifying the 12 V voltage of the power supply, i.e., the battery, to the required voltages (**Figure S1h**). Typically, one will be needed to
modify the voltage to suit the requirements of the camera and one to supply the power to the control unit. The second electrical components are the relays, which will switch the camera and other peripherals to save power. These relays can be either mechanical or based on MOSFETs (**Figure S1i**). One relay is responsible for disconnecting the camera while the second relay plays an important role in allowing remote access to the devices as this relay prevents the WittyPi from automatically shutting down the Raspberry Pi.

**2.2.5 Protective case**

All these components have been installed in an enclosure to protect them from environmental conditions (**Figure 1a and b**). The boxes were perforated to embed a circular photographic filter which acts as protective glass. For the specific development of the Puigcercós and Alhambra de Granada systems, metal boxes were used, which provided very good robustness but some poor reception of the Wi-Fi signal. The anchoring of the boxes in their location varies depending on the characteristics of the
study area. In the case of Puigcercós, concrete bases built ad hoc were used to anchor the boxes (**Figure 1c**), while in Granada they were mainly located in building sites, such as window frames or balconies. In the same way, the solar panels were either supported on the wall or anchored by means of pillars on the concrete bases (**Figure 1d**).

The total cost of the components of each photo module cannot be standardised, as it is controlled by the cost of the camera and lens used. However, the basic components (excluding the camera and lens) usually cost around € 120. The basic connectivity
module components can be purchased for around € 300.

**2.3. Module assembly**

The assembly of these units requires basic electrical knowledge, as they require correct polarity management of the wires and soldering of the devices. In addition, a multimeter is necessary to correctly adjust the output voltages of the DC-DC converters. To activate the solar charge controller the 12 V battery is connected to it. Once the solar charge controller is powered, the solar
panel can be connected to it (connecting it beforehand may damage the charge controller). From the solar charge controller, there are two power lines: one line serves to supply power to the control unit, while the other serves to supply power to the camera. The line that powers the control unit, incorporates a DC-DC converter that transforms the 12 V to the 5 V used by the Raspberry Pi. This DC-DC can be avoided by connecting the 12 V directly to the WittyPi board (from version 3 onwards), but this will result in poorer conversion efficiencies and higher heating of the board (**Figure 2**).

The line that delivers power to the camera also uses a DC-DC converter as the cameras work with a lower voltage (mostly around 7.5 V). This can be identified on the original batteries or on the product datasheet. The camera is connected to the




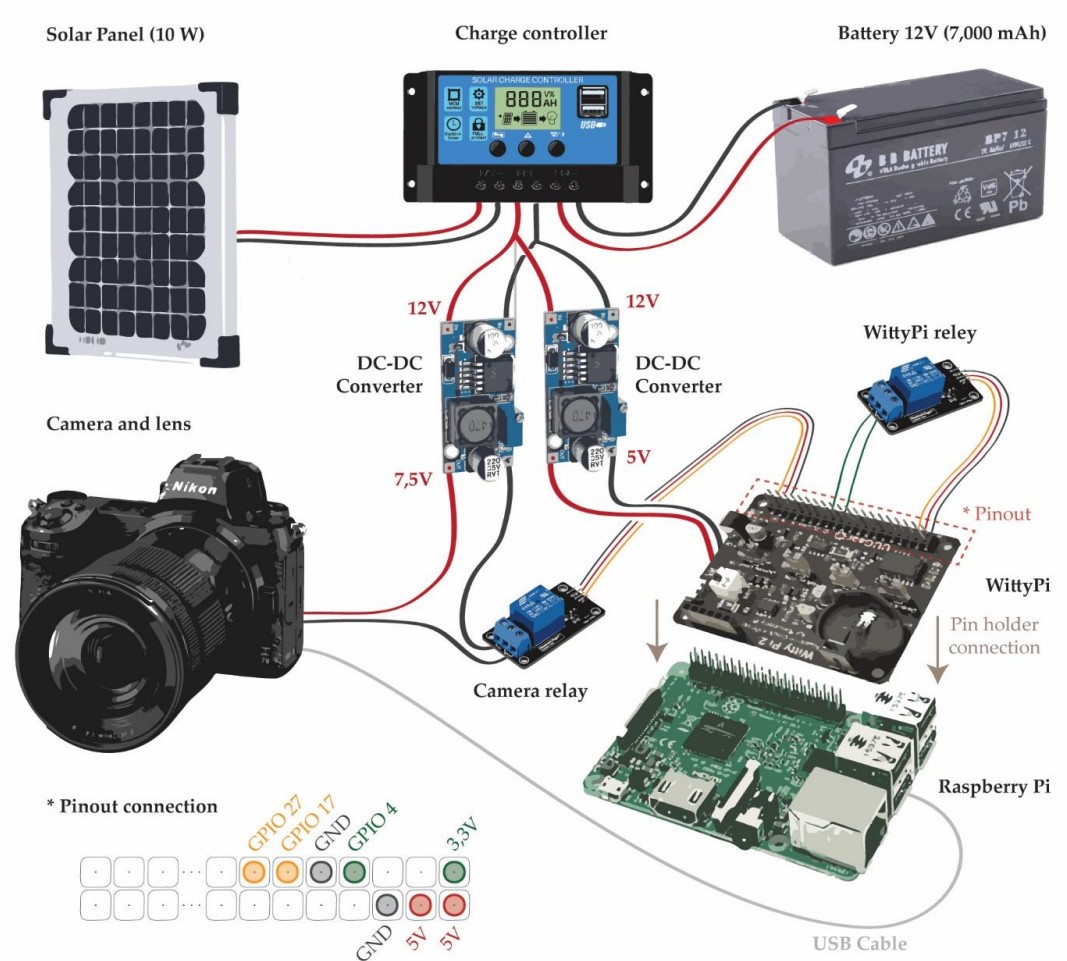

**Figure 2. Installation sketch of the components. The solar panel and the battery are connected to the solar charge controller. Voltage regulators are installed at the output of the controller to adjust the voltage to the required for the camera (7.5V) and the control unit (5V). The relays are controlled by the control unit and allow switching the camera on and off as well as activating the system shutdown. The pinout connection of the control unit is shown in detail in the bottom left of the image.**

power supply using a dummy battery. In addition, the camera power line incorporates a relay to prevent the camera from being turned on continuously. The relay is connected to the GPIO pins of the Raspberry Pi (**Figure 2**).

Finally, the camera is connected to the Raspberry Pi via USB. It is important to note that most modern cameras can also be powered via the USB connection if the output amperage is sufficient for the camera. In this case, it is not necessary to create a power line with the DC-DC converter and the relay. The camera will be activated with the correct voltage when the Raspberry Pi receives power. Finally, the designed system incorporates a relay between GPIO 4 and the 3.3V GPIO. When this relay is closed, the WittyPi board deactivates the automatic shutdown of the equipment, and it remains switched on. This allows active remote connections to be maintained since the system will not shut down automatically.

**2.4 Proposed workflow and software**

A code pipeline has been developed that starts with the photographic capture and ends with the telematic transmission of images while providing a backup on the device's internal memory until it is depleted (**Figure 3**). The workflow comprises of the following steps: a) image capture, b) transfer of images from the camera to the control unit, c) addition of a timestamp to the file name, d) storage of images in a temporary folder, e) uploading of images to a server, and f) transfer of uploaded images

to the control unit's backup (**Figure 3**). Basic information on installing the OS on the Raspberry Pi and the WittyPi software can be found in the supplementary material.


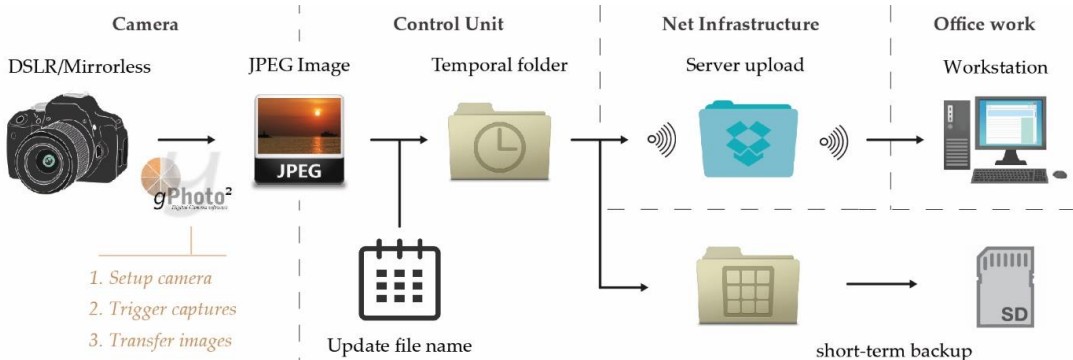

**Figure 3. Illustrative scheme of the workflow implemented in the control unit to communicate with the camera, trigger the camera, send the images remotely, avoiding filling the internal memory of the devices.**

The integration of different devices (i.e., camera, Raspberry Pi, WittyPi, Relays) implies developing adapted codes to integrate them effectively. The use of commercial cameras implies the need to use software to externally control the camera (non-native system) both in terms of communication and power supply. Thus, the Raspberry Pi communicates with commercial cameras
255 via the gPhoto2 framework. The gPhoto2 software and its accompanying libgphoto2 library enable the Raspberry Pi to serve as an advanced control unit for commercial cameras. GPhoto2 is a free, redistributable, and ready-to-use suite of digital camera software applications for Unix systems. The workflow described in **Figure 4** can be reproduced using the codes provided in the first author's GitHub repository. The codes are divided into a main code and auxiliary codes written in Python and bash.

The main code (*main.py*) is in charge of performing the following operations (**Figure 4**):

260  1. Delete files from the camera (avoid corrupted images remaining).
  2. Create/check correct paths.
  3. Remove old files (more than […] days) from "backup folder".
  4. Identify camera.
  5. Set up capture properties and focus camera (if needed).
265  6. Capture images and download images to the unit control.
  7. Change file name by timestamp.
  8. Store it in "temporal folder".
  9. Try to upload to Dropbox server (library: *Dropbox*).
    a. If success -> Move to backup folder.
270    b. If fails -> remain in "temporal folder".

The main.py code is executed by an auxiliary code run.sh, which is in charge of starting the different auxiliary codes and eventually giving the instruction to shut down the system. The log.py code is used to collect a log of the execution of the software as well as the log of the WittyPi board. Both logs are uploaded to the server.

The proposed system includes a mechanism for remote access to the control unit through the use of a relay and the WittyPi
275 control board. Normally, to initiate the shutdown of the system, the WittyPi control board sends the shutdown command to the Raspberry Pi (**Figure 4**). However, this order is not run if a connection between GPIO4 and 3.3 V is established. In order to control this connection, a relay is integrated into the system with an open state. During normal operation, the WittyPi will activate the Raspberry Pi's shutdown command at the end of the run.sh code.

However, the keepalive.sh code activates the relay allowing the connection between GPIO4 and the 3.3 V pinouts (**Figure 2**).
280 In this case, the automatic shutdown is stopped and remote access to the control unit can be established through a remote desktop connection. To revert the situation, the shutdown.sh code is in charge of deactivating the relay that forces the shutdown of the control unit.



**Figure 4. Structured flowchart of the open-source codes distributed in the repository of this publication. In grey, the startup code "run.sh", in orange the main code "main.py", in green the code "logs.py" and in blue, the code "keepalive.py" that avoid the shutdown process.**

## 3 Results

### 3.1 Implementation in real scenarios

The systems presented in this publication have been installed in the study area of Puigcercós (NE Spain) and the study area of Alhambra de Granada (S Spain). Both systems have been installed in front of rockfall active cliffs and have been in operation for over a year (Blanch, 2022). The Puigcercós system comprises five photographic modules (**Figure S4**), which are situated approximately 100 meters from the escarpment and are separated by approximately 30 meters from each other. The connectivity module is installed at the highest point of the escarpment to optimize 4G signal reception (**Figure S3a, b and c**).

This system is located in the same area as the low-cost system used in Blanch et al. (2020), and where the LiDAR that captured the data in Abellán et al. (2009) was placed. The system installed in the Alhambra de Granada is composed of five photographic modules (**Figure S5**) and three different connectivity systems (**Figure S3d**). Due to its urban grid location, the camera geometries in this system are irregular, with the closest camera being 70 meters from the escarpment and the furthest being

**Figure 5. a)** Point cloud obtained with the very high-resolution photogrammetric system installed in Puigcercós (obtained with MEMI workflow). **b)** M3C2 (Lague et al., 2013) comparison of two consecutive days (no expected deformation). **c)** Same comparison than in b) highlighting in red the points with a deformation higher than 0.036 m.

more than 250 meters away. Nevertheless, both systems have a common feature, which is that each camera captures a full view of the escarpment, resulting in 100% coverage in each image (**Figure S4 and S5**). The cameras are configured to take five images in burst mode once daily, which facilitates the application of change-detection enhancement algorithms (Blanch et al., 2021; Feurer and Vinatier, 2018).

In the Puigcercós area we obtain a GSD (Ground Sample Distance) of approximately 2 cmpixel$^{-1}$ for the Canon devices and 1.7 cmpixel$^{-1}$ for the units with the higher resolution Sony cameras. The MEMI workflow (Blanch et al., 2021) has been applied to a stack of four images acquired by each of the five cameras of the photogrammetric system. With this workflow, an RMS reprojection error of 0.23 (0.38 pix) has been obtained. A dense point cloud (after filtering and clipping the scarp) of about 15 Mio points has been generated with a mean point surface density of 6,000 pointsm² using Agisoft Metashape software with the highest quality setting. The model has been scaled with virtual GCPs (Ground Control Points) extracted from a terrestrial

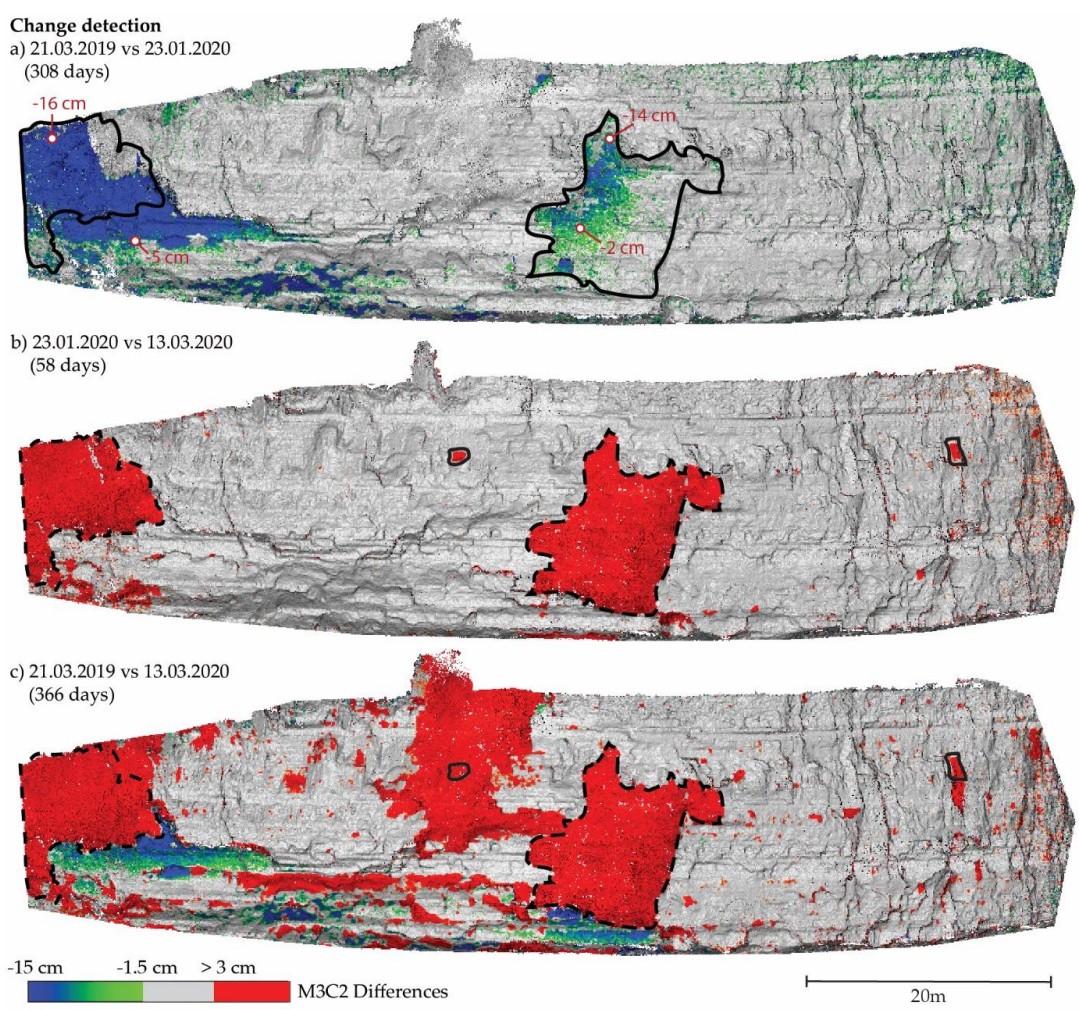

**Figure 6. Examples of pre-failure deformations and rockfalls detected. a) Accumulated pre-failure deformation of two active blocks after 308 days. The comparison has been done with the M3C2 algorithm and the MEMI workflow. b) Change detection obtained 58 days after the detection of pre-failure deformation shown in a), with the two detached blocks identified in red. c) Change detection after approximately one year. The red clusters identify rockfalls, while the green-blue areas indicate pre-failure deformation. For all results 0.015 m has been used as level of detection.**

LiDAR model of the same study area. **Figure 5a** shows the photogrammetric model obtained on the Puigcercós cliff, while

**Figure S6,** in supplementary material, showcases the 3D model generated in the Alhambra study area. Links to an online view of a subsample models are also provided in the **Appendix A** (Puigcercós cliff, Spain) and **Appendix B** (Alhambra de Granada cliff, Spain).

In order to estimate the instrumental error of the photogrammetric system and estimate the minimum change detection value, two captures obtained in Puigcercós between 2020.07.03 and 2020.07.05 were compared. Since there has been no significant

change in the analysis period, the result of the comparison should tend to zero (**Figure 5b**). A mean value of the comparison of 0.08 mm and a standard deviation of 0.018 m has been obtained using the M3C2 algorithm for point cloud change detection (Lague et al., 2013). Although this result is highly dependent on the algorithm, as will be discussed below, this result allows us to define a change detection threshold of only 0.036 m, using the methodology described in Abellán et al., (2009) that describe the detection threshold as two times the standard deviation of a model comparison without deformation. If we apply

the same methodology but on a crop in the central area of the escarpment (20%), the standard deviation obtained decreases to only 0.007 m, which corresponds to a change detection threshold of only 0.014 m.

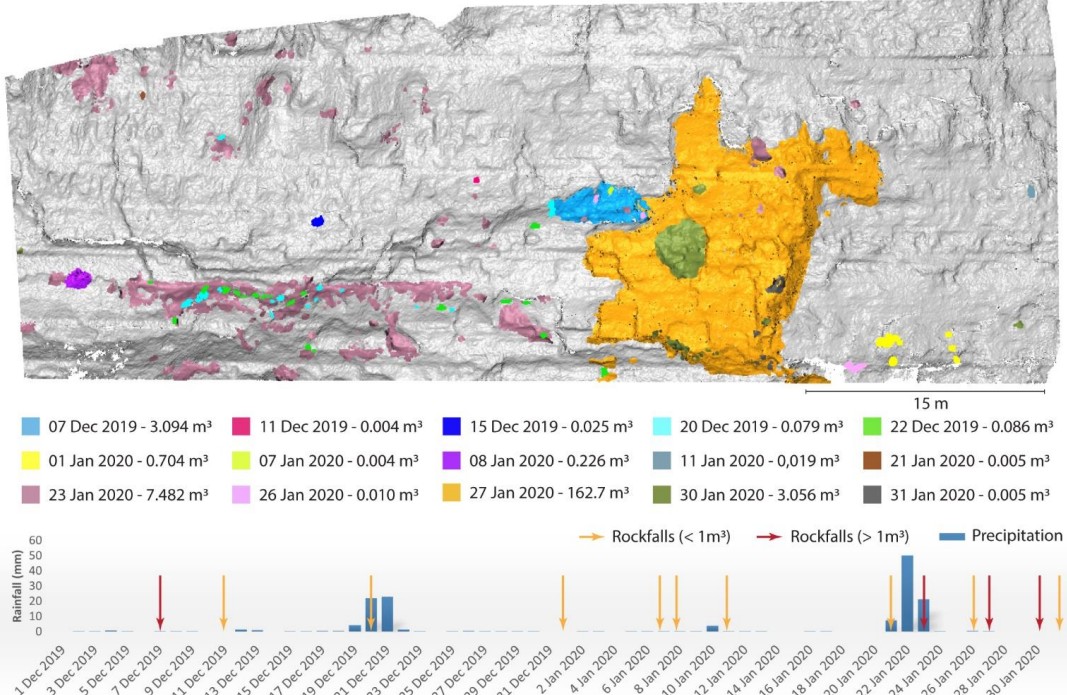

**Figure 7. Rockfalls occurred on the Puigcercós cliff (NE, Spain) between December 1st and January 31st. Each color corresponds to a different detection date, and the total volume of rockfall per day is calculated. The graph displays the precipitation levels in the area and the days with rockfall activity. Red arrows indicate a total volume detached greater than 1m³, while orange arrows indicate a volume less than 1m³.**

**Figure 5c** highlights in red the points that exceed the change detection threshold of 0.04 m in the comparison between the two days period without deformation. It is noteworthy that most of the points with higher deformation are located at the margins of the model. Both the points exceeding the threshold found at the edges of the model, and those found in more central areas (these more scattered), are associated with fractures, edges, and shadows of the 3D model. The points shown in red represent 0.24% of the total point cloud, so 99.76% of the points in the M3C2 comparison fall within the deformation range of -0.036 m to +0.036 m for a comparison without deformation (consecutive days without rockfall activity).

### 3.2 Contribution to process understanding: detecting rockfalls and pre-failure deformation.

With the fixed time-lapse photogrammetric systems, results can be obtained that benefit from the high frequency of data acquisition. The comparison of point clouds with 308 days' time lapse performed with the MEMI workflow (Blanch et al., 2021) allowed detecting pre-failure deformations in two active zones of the Puigcercós cliff (**Figure 6**). The observed deformation increases along the vertical axis of the mobilized blocks reaching deformation values of 0.16 m in the left block and 0.15 m in the right block, thus the process can be characterized as rock toppling (Hungr et al., 2014). Although in the previous section we calculated the theorical limit of detection at 0.04 m, a change detection threshold of only 0.015 m was used to identify the deformation in the figure. The use of this threshold is supported by two observations. The first corresponds to the identification of a standard deviation of change-detection of only 0.007 m in the central 20% of the escarpment, the second observation is based on the homogeneity and density of the points that generate the deformation clusters in large volumes. For this reason, **Figure 6a** shows useful results obtained with a change-detection threshold of only 0.015 m, which, although it shows a greater amount of noise especially at the margins (far from the central 20% of the escarpment), allows the identification of the pre-failure blocks.

The comparison covering a period of 58 days after the detection of two active zones with pre-failure deformation revealed the detachment of two rockfalls of great magnitude (more than 100 m³) (**Figure 7b**). It is noticeable that in the central rockfall the detached zone overlaps with the deformation detected in **Figure 6a**, while at the rockfall on the far right the detached zone does not cover the entire pre-failure deformation area detected in Figure 8a. For this reason, in later comparisons (**Figure 6c**) part of the deformation is still detected at the left end of the figure (part of the mobilized block remains active).





**Figure 6c** shows a comparison of the detection of changes in the Puigcercós cliff after one year (366 days), covering the two time-lapses shown in **Figures 6a** and 6**b**. Therefore, the figure shows the rockfalls detected in **Figure 6b** but as part of larger rockfall, which makes it impossible to individualise them and leads to an over-interpretation of the rockfall volumes of each event. In addition, one of the small rockfalls detected in **Figure 6b** is completely masked by a new larger rockfall in **Figure 6c**, which results in ignoring part of the rockfall activity of the cliff. A problem that is solved by fixed photogrammetric systems that allow daily or sub-daily frequency monitoring.

Last, **Figure 7** shows another example of monitoring the Puigcercós cliff, which benefits from high temporal resolution data acquisition provided by the fixed time-lapse photogrammetric systems described in this publication. The figure shows a daily analysis carried out over a period of 2 months (December 2019 - January 2020), where rock falls are identified, and the volumes of all material detached per day are included. A threshold of 0.05 m has been used to identify positive deformations (rockfalls). As in the previous **Figure 6**, overlapping rockfalls are observed in the areas of greatest activity. In addition, it is observed that this overlapping occurs closely in time (27-30-31 January 2020), allowing us to infer a triggering sequence. The largest detected volume corresponds to the 162 m$^3$ rock fall that occurred on January 27, 2020, while the smallest detected volume is 0.004 m$^3$, corresponding to a block approximately 20x20 cm and 10 cm deep. Finally, a graph is included with the precipitation recorded in the village of Tremp (NE Spain) located 4 km from the Puigcercós cliff (XQ-XEMA, Meteorological Service of Catalonia). The graph also displays the days with rockfall activity, categorized by magnitudes greater than and less than 1m$^3$.

## 4 Discussion

### 4.1 Overall approach

The results obtained in the rockfall monitoring in Puigcerós (NE Spain) (**Figure 6 and 7**) demonstrate that the photogrammetric system presented in this publication is useful for 3D change-detection using high-end cameras and is particularly appropriate for those processes that require a high temporal frequency of data acquisition while maintaining good accuracy. The systems installed in Puigcercós cliff and Alhambra de Granada have been in operation for more than one year, requiring some maintenance to keep them operational. In some cases, continuity of data acquisition could not be guaranteed. However, thanks to the use of independent systems, even with the failure of two photographic modules the photogrammetric system captured images from three different locations guaranteeing 3D results. This allows us to obtain, as a whole, a more robust system than installations based on stereo cameras.

Maintenance has occurred for a variety of reasons, but there are two main limitations of this type of systems. The first relates to the specific design of the systems and their ability to withstand a severe climate with precipitation, high humidity and a very wide temperature range. The construction of the boxes are weak points because they allow humidity to enter, which can seriously damage the electronic systems (**Figure 8a**). Although many efforts to waterproof the electronics were taken, it was not possible to completely isolate the systems. Furthermore, the reached temperatures inside the metal boxes in Puigcercós were high (reaching 60 degrees Celsius). Thus, solutions such as silicone sealant, power tape or thermal adhesives were not valid as they can soften and lose their firmness and properties over time. Other methods such as encapsulation in epoxy resin or insulation with high resistance polymer had been tried, but the operation is very difficult because the electronic components have elements that cannot be submerged or covered by resins (e.g., microSD card or USB connectors).

One way to improve the insulation of these enclosures is to replace the circular photographic filter that is embedded in the enclosure. Due to weather conditions, the filter glass was no longer fixed to the metal chassis that held it in place, allowing the glass to rotate freely and consequently allowing humidity to enter. Other alternatives such as adding a larger and thicker glass window to the enclosure should allow for greater strength and insulation. In addition, the use of metal boxes hindered the reception of the Wi-Fi signal (Faraday cage effect), forcing the need of an extra hole for the external antenna. In addition, it should be noted that not all pictures are systematically usable for monitoring, unlike other instrumentation, the robustness of the results is conditioned both by the time of day (image acquisition only during daylight hours) and by weather conditions (**Figure 8b**). Thus, it is not possible to obtain images during periods of heavy rain, fog, or direct sunlight/flairs on the lens. To try to mitigate the negative effects caused by drops of water on the glass or light reflections, a hood has been added to the system (**Figure 8c**). Although the results have not been noticeably better, given the low cost of the improvement, it is a change to be considered.

### 4.2 Hardware and software

Another limitation of the system has been the integration of all the hardware from different brands that comprise the system. These limitations range from the failure of the electronic components to the need to rewrite the code because some cameras are not fully compatible with the gPhoto2 library used to connect the camera to the control unit.

The main electrical installation can be seen in detail in **Figure 2**. However, it was decided to replace the mechanical relays (electromagnets) for static relays SSR (solid state relays). The mechanical relays have caused problems in the different



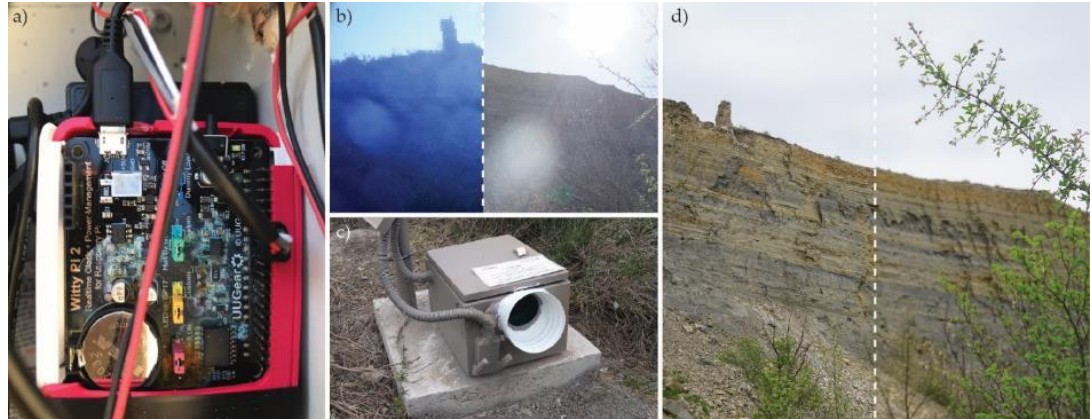

**Figure 8. Limitations of the high-quality system: a) Rust on the control unit due to water infiltration; b) Image defects due to not incorporating a lens hood. Left: Water drops in the filter glass. Right: lens flare due to the sun's rays; c) Incorporation of the lens hood to avoid the defects shown in b) and the external Wi-Fi antenna; d) Problem due to automatic focus. Left: image with the focus on the cliff. Right: image with the focus on the vegetation in the foreground.**

installations due to either the quality of the devices or climatic conditions. For this reason, they have been replaced by MOSWITCH-SPDT relays of the UUGear brand. These new relays no longer depend on a mechanical component that is 395 activated by the passage of current (electromagnet) that activates the switch as in mechanical relays. In contrast, the SSR relays are activated by the state of activation of the MOSFET transistors (metal-oxide-semiconductor field-effect transistor) incorporated in the device.

The most distinctive aspect of this high-resolution system lies in its camera selection. The chosen camera significantly impacts the cost of the system, the communication protocol with the control unit, and even the electrical system. Based on the results 400 obtained using the different systems, **Table 1** gives a short summary of critical factors in order to optimize the results.

On DSLR cameras the focus position can be fixed externally on the lens. However, the new mirrorless cameras incorporate much more electronics than their predecessors, and many of them perform automatic focus travel at start-up so that the last focus position is not memorised even when manual focus is selected. For this reason, some of the systems created in this research had to activate the automatic focus of the cameras, which led to higher battery consumption, a longer delay in the 405 acquisition of photos and, most seriously, an error in obtaining the focus (**Figure 8d**).

In the same way, choosing cameras with full compatibility with the gPhoto2 library makes it possible to generalise the image acquisition codes. In this study, the Sony camera was not able to execute the burst shooting command. The camera could only be controlled by the "shoot and download" command, which made it difficult to obtain simultaneous bursts with the other photographic units.

The possibility of powering the new mirrorless cameras via USB cable directly from the control unit makes it possible to eliminate the need for a power supply line, a relay and a DC-DC converter. However, it must be ensured that the current supplied by the USB port of the Raspberry Pi is sufficient to power the camera. Last, the cameras with the most electronics are the ones that have reported the most difficulties in the field, as well as incompatibilities and malfunctions (camera freezing or corrupted files), so the choice of the robust DSLR can still be a winning point.

**Table 1. Cameras installed in the different systems and specification of the most important features.**

| Camera model | Sensor size | Sensor resolution | Camera type | Focus memory | gPhoto2 functions | USB Powered |
|---|---|---|---|---|---|---|
| *Nikon D610* | Full Frame | 24 MPx | DSLR | n/a | Full | No |
| *Nikon Z5* | Full Frame | 24 MPx | Mirrorless | Yes | Full | Yes |
| *Canon 77D* | APS-C | 24 MPx | DSLR | n/a | Full | No |
| *Sony α7 II* | Full Frame | 42 MPx | Mirrorless | No | Limited | No |


### 4.3 Applications in real scenarios

The high-resolution system allows high quality change detection models, this result is consistent with other authors such as
Kromer et al., (2019) or Kneib et al., (2022) who have already demonstrated the potential of using automatic multi-camera photogrammetric systems for natural hazard monitoring. The theoretical detection threshold obtained with the high-resolution cameras (0.04 m) is 50% lower than those obtained with the low-cost systems (Blanch et al., 2020). Considering the low-cost systems shown in Blanch et. al., (2020) were located at a shorter distance from the Puigcercós escarpment (60 m), the real differences in change detection thresholds would have been much greater if they had been located at the same distance, with
an increase in the detection threshold of 3 to 4 times. Given that the captured images have 100% coverage of the area of interest and thus have high overlap, the obtained model was robust and showed better model edge coherence compared to very low-cost devices. Moreover, the results were achieved by the use of cameras with different characteristics (sensor size and number of different pixels). It can be assumed that if all systems used cameras with the highest quality (e.g., all cameras with 45 MPx, and same prime lens), the quality of the model could have been much higher, too.

One of the main advantages of using fixed systems is the possibility of acquiring data with high temporal frequency. When comparisons are made using long data intervals, accurate but incomplete information is obtained. On that way, **Figures 6** and **7** highlight the importance of working with high temporal resolutions to obtain accurate rockfall inventories. For example, the four rock falls detected in **Figure 6b** would have been identified as major rock falls if the change detection analysis is performed on an annual basis (**Figure 6c**). Additionally, **Figures 6** and **7** show how the monitored sequence of rockfalls varies
based on the frequency of monitoring. While in **Figure 6c** a single large rockfall is identified, thanks to the daily analysis in **Figure 7** it is possible to identify that the largest rockfall area was the result of different rockfalls occurred on 7th December 2019, 23rd, 27th, 30th, and 31st January 2020. Thus, long time-lapse comparisons generate distorted inventories that do not respond to the natural behaviour of the escarpment (Williams et al., 2018).

This sampling effect is observed in both large and small rockfalls, favouring, for example, the rollover effect in the magnitude-
frequency curves (Van Veen et al., 2017). Moreover, high-frequency monitoring may allow for analyses to assess the relationship between weather conditions and rockfall activity. Although this publication did not focus on this aspect, **Figure 7** suggests a potential correspondence between precipitation and rockfall activity, as the episodes with the highest detached volume (23rd and 27th January) occurred during a precipitation episode, which may corroborate the relationship between weather conditions and rockfall shown in similar studies (Birien & Gauthier, 2023). However, the purpose of **Figure 7** is only
to demonstrate that the presented photogrammetric systems can be a cost-effective and useful tool to accurately obtain these relationships, as such correlations need to be done with longer time series.

In this study, the standard deviation of the comparison has been used as a quality metric for evaluate the change-detection result, and in order to compare with the LiDAR results published in Abellán et al., (2010). However, Blanch et al., 2021 show results obtained with the same photogrammetric system considering the error propagation, and obtaining the accuracies in X,
Y and Z as a function of the location in the 3D model (James et al., 2017). The values published in Blanch et al., 2021, which also allow obtaining change detection thresholds, are consistent with those obtained in this work. Abellán et al., (2010) performed two simultaneous LiDAR captures on the Puigcercós site, with the LiDAR device located in the same place, so the results obtained in both studies can be compared. However, in terms of metrics, it should be noted that the algorithms for change detection are quite different. And in the case of the M3C2 algorithm (used in this study) the metrics are highly
dependent on the parameters used for comparison.

Abellán et al., (2010) obtain a standard deviation of 0.0168 m, highlighting that the artefacts occur at the margins of the occluded parts. This standard deviation value allows to define a change detection threshold of 0.0336 m (Abellán et al., 2009). This value can be directly compared with the result of 0.0360 m obtained in this publication. In addition, the distribution of errors is similar in the two works, highlighting the appearance of errors in shadow zones and fractures of the cliff. (**Figure 5**).
Furthermore, the results obtained are consistent, in terms of change-detection model, with those obtained under different conditions by Kromer et al., (2019).

Finally, it should be noted that due to the number of error points over the threshold (0.24% of total points), its distribution (edge effects and shadows) and their composition (density and clustering) it is possible to use change-detection thresholds below the theorical value. This is supported by the distribution of the error, which instead of being random throughout the 3D
model (as background noise in LiDAR) is concentrated at the extremes of the model and in the shadow zones (associated with fracture zones). Similar works where pre-failure deformation is identified also use detection thresholds below the theoretical one, as for example Royán et al., (2014) apply a NN (near neighbours) moving median to the change-detection result. This strategy allows them to smooth the background noise, obtaining lower standard deviations and consequently lower thresholds. In the example shown in **Figure 6**, the blocks present a large deformation so that it is easily distinguishable from the
background noise, however, the application of NN filters may allow to identify areas with incipient precursor deformation.

This threshold reduction will become increasingly noticeable as AI (artificial intelligence) elements are used to improve the identification of true clusters that correspond to rockfalls. Thus, just as we have seen an increase in algorithms that allow for





improved 3D modelling (e.g., Blanch et al., 2020; Feurer and Vinatier, 2018), it is likely to be possible in the future to see deep learning applications that reduce these detection thresholds without modifying the photogrammetric acquisition systems;
allowing to improve change detection processes significantly.

**5 Conclusions**

Two high-resolution photogrammetric systems have been designed, constructed, and installed in the escarpment of Puigcercós (N Spain) and the Alhambra de Granada (S Spain), comprising five photographic modules and different wireless transmission system for image transfer. These modules use high-end DSLR and mirrorless cameras and are controlled by the gPhoto2 library
and a Raspberry Pi. The acquired images, in combination with algorithms that facilitate the creation of improved 3D models, offer change detection performance comparable to LiDAR. The system presented has been used to successfully monitor rockfalls, allowing comparison of 3D change detection models with the same accuracy as previous work using LiDAR. Thus, the results obtained demonstrate the usefulness of these systems for monitoring natural hazards. In addition, our findings support that a high frequency of monitoring yields a more accurate rockfall inventory, and consequently a more reliable
characterisation of the behaviour of the active cliffs.

**Appendix A:** High-resolution 3D photogrammetric model (Puigcercós cliff, Spain): https://skfb.ly/o7QvV

**Appendix B:** High-resolution 3D photogrammetric model (Alhambra de Granada cliff, Spain): https://skfb.ly/o7QEX

**Code availability:** all the necessary codes supporting the contents of this publication are available in the first author digital repository: https://github.com/xabierblanch/DSLR-System.

**Author Contributions: XB**: Conceptualization, methodology, software, investigation, writing-original, figures; **MG:** conceptualization, writing - review and editing, supervision; **AE:** conceptualization, methodology, writing - review and editing, supervision; **AA:** conceptualization, writing - review and editing, supervision.

**Competing interest:** The authors declare that they have no conflict of interest. The use of brand names in the manuscript has been made without any commercial relationship.

**Disclaimer:** All data, figures and codes provided in this publication are published under open-source license. The content of this publication is adapted from the Doctoral Thesis of the first author (Blanch, 2022), that is available at the following link: http://hdl.handle.net/2445/189157

**Acknowledgments:** The authors would like to thank Origens UNESCO Global Geopark for granting permission to work on Puigcercós rock cliff, the Patronato de la Alhambra y el Generalife for granting permission to work on Tajo de San Pedro cliff,
to the institutions and private buildings that have accepted and cared for photogrammetric systems on their properties, to the colleagues of the University of Granada who have assisted us in the fieldwork, the doctoral thesis examining board for their comments that have improved this publication and the reviewers and the editor for the valuable comments and suggestions that contributed to the improvement of the present manuscript.

**Financial support:** The presented study was supported by the PROMONTEC Project (CGL2017-84720-R) funded by the
Ministry of Science, Innovation and Universities (MICINN-FEDER). The first author (X. Blanch) was supported by an APIF grant funded by the University of Barcelona. Anette Eltner was funded by the DFG (EL 926/3-1).

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
