# Peer review of "Fixed photogrammetric systems for natural hazard monitoring with high spatio-temporal resolution"

_Natural Hazards and Earth System Sciences, 2023_

## Referee Comment (RC2)

[referee-annotated manuscript omitted]

---

## Author Response (AR1)

Dear reviewers.

Thank you for taking the time to provide your constructive feedback on our manuscript and for making it promptly. Your comments certainly have improved the quality of our manuscript and we are grateful for the minor revisions and typos highlighted in your comments. All the mentioned points have been modified, corrected, or expanded in the reviewed version of the manuscript.

**General comment**

In this revised version of the manuscript, I have made the following relevant changes:

- At the request of the two reviewers, I have added a paragraph (line 305) explaining the photogrammetric processing used for this research (MEMI, Blanch et al., 2021).
- At the request of R2 I have modified the figures: Figure 2 and Figure 5. And at the request of R1 I have added Figure S3 and S5.
- I have extended the conclusions and modified the citations to adapt them to the journal format (including a new preprint).

In addition, I have added/modified all the comments you have made specifically in your review.

**Reviewer 1**

*Line 285-295: It could be useful for the reader to have maps of the study areas with the location of the camera's network used for monitoring because the mere reference to previous work (Blanch et al., 2020) is not sufficient for the manuscript to have its own independence in understanding the carried-out study.*

We have added two figures in the supplementary material with camera locations (Figure S3 and S5)

*Line 299: Please briefly explain the workflow and acronym. The citation alone is not sufficient to ensure an understanding of the work done independently of previous work. In this way, the reader can better understand the study.*

As I indicated in the general comment, on line 305 onwards I include a brief explanation of the photogrammetric workflow used (MEMI, Blanch et al 2021).

*Line 303: Where are the GCPs located in the study area (a figure could be added about this)? Is it a network of fixed points? Did you use also independent Check Points (CPs) in the photogrammetric workflow? They are fundamental to understanding the errors and accuracy of the point cloud.*

In line 315 we have explained how we use virtual PCMs and why we have used them in this study.

*Has the problem of co-registration of point clouds been considered in the multi-temporal 3D surveys? This problem could affect the measure of the minimum change detection value between multi-temporal point clouds.*

Answered in the public comment.

**Typing errors**

Line 298: 2 cm/pixel or 2 cm pixel-1  - done

Line 301: error of 0.23 m?? (units of measurement are missing) - not necessary

Line 302: what does "Mio points" stand for? - done

Line 302: 6000 points/m2 or 6000 points m-2  - done

**Reviewer 2**

*Chapter 2: ...What I am missing here are details on the processing of the images in the next steps. Surely, the aim of the authors is to provide instructions for setting up a photogrammetric system, not going into details on the generation of digital topography from the images acquired. However, as the Chapter 3 looks at the accuracies of the models that can be attained, it would be nice to at least provide some short paragraph on the workflow and settings used for the generation of 3D models.*

As I indicated in the general comment, on line 305 onwards I include a brief explanation of the photogrammetric workflow used (MEMI, Blanch et al 2021).

*The same applies for the M3C2 algorithm; here the authors point to the fact that the settings have a crucial influence on the results – providing those details for the examples would be very much appreciated.*

In line 328 we have included the parameters used for the M3C2.

*L212-214 and L239-244: This only becomes clear later in the text. Maybe the authors could also indicate at this point that the idea of this second relay is to trigger it by running a code in the case that the system should not shut down automatically.*

Modified. We have changed the text in lines 210-215 and again in 245-250.

*L439-455: As the paper is rather focusing on the technical aspects of the setup, this section comes a bit as a surprise. While I think some of the points are valid, they would require an in-depth analysis. I would recommend leaving out this section here and focus on the discussion*

Agree. We remove the first paragraph, but we keep the discussion about the sampling effect (line 457-461) because we consider that is relevant and it is one of the benefits of this type of systems (4D). Regarding the second paragraph, we remove part of the redundant information and we clarify the text, focusing on the results that we obtained.

*PDF Document*

I have modified and fixed all the comments that were indicated in the .PDF document.

Thank you very much for your time, and I hope that this new version of the manuscript will be satisfactory to you.

The authors